# Driving Paediatric Vaccine Recovery in Europe

**DOI:** 10.3390/vaccines11010184

**Published:** 2023-01-15

**Authors:** Claire Alexander, Mariel Cabrera, Michael Moore, Marta Lomazzi

**Affiliations:** 1Brunswick Group, Avenue des Arts 27, 1040 Brussels, Belgium; 2World Federation of Public Health Associations (WFPHA), ch des Mines 9, 1202 Geneva, Switzerland; 3Institute of Global Health, University of Geneva, ch des Mines 9, 1202 Geneva, Switzerland

**Keywords:** vaccination coverage, vaccine uptake, vaccine hesitancy, misinformation, vaccine accessibility, COVID-19, Ukraine crisis

## Abstract

Background: Childhood vaccination coverage has increased throughout Europe in recent decades. However, challenges persist in many areas within the European Union (EU), resulting in declining coverage rates in many countries in the period between 2010 and 2021. This general trend requires increased efforts to combat barriers around vaccination uptake. Thus, this article aims to summarise key learnings and trends in paediatric vaccination within the EU, with a focus on current challenges and enablers. Methods: Methodology is based on analysis of primary data, mainly vaccination coverage rates, as well as review and analysis of the select relevant literature, including peer-reviewed articles, academic research papers, official reports, policies, and other publicly available sources. Results: For all vaccines assessed (DTP 1st dose, DTP 3rd dose, Hib3, HepB3, measles 1st dose, measles 2nd dose, and polio 3rd dose), a high degree of variation and fluctuation in coverage can be observed. There is a general trend of declining coverage in 2019 compared to 2010, with lower performing countries, such as Romania and Austria, showing increasingly severe coverage fluctuations between the years examined across the analysed vaccines. Conclusions: Evidence suggests that increasing both accessibility and information regarding vaccines are key enablers to vaccination uptake. Moreover, given the current challenges the EU is facing, crisis preparedness plans are pertinent to ensure immunity gaps do not further exacerbate the disruption of vaccination systems.

## 1. Introduction

Vaccination is one of the most powerful and cost-effective tools in the history of public health [1], with important health, economic, and social benefits [2]. Effective paediatric vaccination programmes protect both children and adults against sometimes life-threatening diseases [3]. When large portions of populations are vaccinated, “herd immunity” or “herd protection” reduces the transmission of vaccine-preventable infectious diseases [4]. This leads to the in-direct protection of unvaccinated children and adults, including those not eligible for vaccination (e.g., immunocompromised, allergic, etc.) and hard to reach or excluded populations [4]. Paediatric vaccination contributes to: (1) reducing mortality and morbidity globally, (2) reducing disease incidence, and (3) health protection [2,3]. In short, it is a key driver in disease prevention.

Within Europe, childhood vaccination coverage has increased in recent decades, with several countries managing to achieve the 95% coverage goal within the last ten years. However, coverage rates in many countries declined in the period between 2010 and 2021 [5]. As a result, several European countries have experienced unprecedented outbreaks of vaccine-preventable diseases [5]. Most notably, there have been major measles outbreaks; cases increased between 2017 and 2018 [6], with 74 deaths in 2018 due to complications of the disease [7]. Low and declining vaccination coverage rates (VCRs) in non-EU countries bordering the EU are also a cause for concern, as outbreaks in these countries could spread to the EU [8,9]. Moreover, the ongoing pandemic, alongside the current conflict in Ukraine, further exacerbates the threat of outbreaks of vaccine-preventable disease in the EU through the disruption of routine immunisation programmes [10,11,12]. The largest sustained decline in paediatric vaccinations over the past 30 years has been recorded in official data published by the World Health Organization (WHO) and United Nations Children’s Fund (UNICEF) [11]. Globally, data from the first four months of 2020 indicated a decline in Diphtheria Tetanus Pertussis (DTP) coverage, which is generally considered a reliable marker of vaccination coverage [10]. VCRs declined in every region, increasing the immunisation gap, and resulting in avoidable outbreaks of vaccine-preventable diseases (VPDs) [10,11]. All these factors, alongside regular fluctuations in vaccination coverage levels, draw a scenario of enhanced outbreak risks in the EU.

Decrease in VCRs are a concern; it signals a lack of reliability and resilience of vaccination programmes. Additionally, it is associated with an increase in un- and under-vaccinated individuals, with a higher likelihood of VPD outbreaks. Under-vaccinated individuals are defined in this article as those who have not received all recommended doses for a particular vaccine. In this challenging context, robust and resilient vaccination systems across the EU are crucial to protect the population from preventable illnesses and death due to vaccine-preventable diseases (VPDs), as well as ensuring recovery from any future crises. Through the analyses of the changes in vaccination coverage of these vaccines in the EU, alongside an assessment of the impact of policy interventions on VCRs, this article denotes key levers and barriers around vaccination uptake to maintain or increase coverage rates.

## 2. Materials and Methods

### 2.1. Vaccines Analysed

The vaccines analysed within this article were chosen based on current WHO indicators and recommendations, which are DTP 1st dose, DTP 3rd dose, Haemophilus influenzae type b (Hib3), Hepatitis B 3rd dose (HEPB3), measles 1st dose, measles 2nd dose, and polio 3rd dose. Due to the absence of available data or recommendations in paediatric vaccination programmes within EU countries, the pneumococcal and Bacillus Calmette–Guérin (BCG) vaccines were excluded. It is important to note that, at the time of analysis, there was a lack of data for HepB3 vaccination coverage for Slovenia, 2020, as well as measles 2nd dose vaccination coverage for Finland, 2010, 2014, and 2015; Luxembourg, 2010; Cyprus, 2020.

Analysis of primary data extracted from an externally available dataset was conducted in July 2022. This included vaccination coverage data from all 27 EU Member States between 2010 and 2021. Coverage rates per vaccine are framed in the context of the 95% vaccination coverage threshold needed to prevent vaccine-preventable outbreaks and reach “herd immunity”, as recommended by WHO [13,14]. Unless otherwise specified, all vaccination coverage data in this review is based on the WHO/UNICEF/Joint Estimates of National Immunisation Coverage (WUENIC) data [15]. When WUENIC data was unavailable, other comparable data was utilised (e.g., based on other available WHO data or official national datasets).

### 2.2. Literature Analysis

A targeted analysis of the select relevant literature was conducted, including peer-reviewed articles, academic research papers, official reports, policies, and other publicly available sources. This included research into trends in vaccination coverage and vaccination policies across the EU and in select Member States, including Austria, Bulgaria, Germany, Italy, and Romania. Key words included vaccination, immunisation, vaccination programmes, vaccination coverage rates, vaccination policies, vaccines, vaccine schedule, infectious disease outbreak, measles outbreak, disruption of vaccination services, Ukraine refugee crisis, and conflict in Ukraine. This literature analysis serves to contextualise and complement the data analysis to heighten understanding of trends, patterns, and the relationship between policy interventions and vaccination coverage rates.

## 3. Results

To assess the state of the changes in vaccination coverage in the EU over time, a time series analysis of vaccination coverage rates was conducted for the period from 2010 to 2021 based on the WHO vaccination indicators of DTP 1st dose, DTP 3rd dose, Hib3, HepB3, measles 1st dose, measles 2nd dose, and polio 3rd dose.

VCRs vary across the EU over time and across different vaccines, countries, and regions. For all vaccines assessed, a high degree of variation and fluctuation in vaccination coverage rates in the EU over the period from 2010 to 2021 can be observed. There is an overarching trend of declining vaccination coverage across vaccines in 2019 compared to 2010. Lower performing countries, such as Austria or Romania, show more frequent or more severe fluctuations across vaccines. In contrast, higher performing countries tend to have little or no fluctuation in coverage.

### 3.1. DTP Vaccine

Whilst coverage for DTP 1st dose was higher in 2019 than in 2010 for Croatia, Denmark, Italy and Malta, coverage in 2019 had fallen compared to 2010 for Austria, Cyprus, Estonia, Finland, Germany, Lithuania, The Netherlands, and Sweden (Figure 1). In 2010, Denmark was the only country with a coverage rate below the 95% target threshold. In 2019, this coverage increased to 97% in Denmark while declining in both Austria and Estonia to below the target threshold. The decline is particularly steep in Estonia, where coverage decreased from 96% in 2010 to 92% in 2019.

The coverage rate dropped in Belgium, Bulgaria, Italy, Lithuania, and Romania in 2020 compared to 2019, bringing Bulgaria and Italy below the target threshold to 94% (Figure 1). Coverage was lower in 2021 compared to 2020 in Bulgaria, Czech Republic, Estonia, Latvia, Lithuania, Slovakia, Slovenia, and Spain.

For DTP 3rd dose (Figure 2), coverage was higher in 2019 than in 2010 for Denmark, Italy, Latvia, Malta, and Portugal. For Austria, Bulgaria, Croatia, Cyprus, Czech Republic, Estonia, Finland, France, Germany, Lithuania, The Netherlands, Poland, Romania, Slovakia, Slovenia, and Spain, coverage was lower in 2019 compared to 2010. Declines in coverage for DTP 3rd dose is more marked than for DTP 1st dose. Thirteen of the twenty-seven EU Member States fell below the 95% target in 2019 compared to 10 in 2010. Some countries were more affected in one particular year, such as Belgium and Sweden—where coverage fell in 2020 but was recovered to their previous rate in 2021.

There was a decline in coverage in 2021; coverage was lower in 2021 than in 2020 for thirteen countries (Austria, Bulgaria, Croatia, Czech Republic, Estonia, Finland, Germany, Italy, Latvia, Lithuania, Romania, Slovenia, and Spain).

### 3.2. Hib3 Vaccine

Hib3 coverage (Figure 3) was lower in 2019 than in 2010 in sixteen countries (Austria, Belgium, Croatia, Cyprus, Czech Republic, Estonia, Finland, France, Germany, Lithuania, The Netherlands, Poland, Slovakia, Slovenia, Spain, and Sweden). Between 2019 and 2020, VCRs remained consistent in most EU countries. It dropped in six countries (Bulgaria, Croatia, Italy, Lithuania, Poland, and Romania), and increased only in Spain. Hib3 coverage rates remained constant between 2019 and 2021 in eleven countries (Austria, Belgium, Cyprus, Denmark, France, Greece, Hungary, Luxembourg, The Netherlands, Portugal, and Slovakia). For twelve countries, VCRs declined in 2021 compared to 2020 (Bulgaria, Croatia, Czech Republic, Estonia, Finland, Germany, Ireland, Latvia, Lithuania, Romania, Slovenia, and Spain). They increased in 2021 only for Malta and Sweden.

### 3.3. HepB3 Vaccine

HepB3 vaccination coverage followed a similar trend in decline (Figure 4), with VCRs lower in 2019 than in 2010 for thirteen countries (Austria, Bulgaria, Croatia, Cyprus, Czech Republic, Estonia, Germany, Italy, Lithuania, Poland, Romania, Slovakia, and Spain). A large decline was seen in Romania, where coverage fell from 98% in 2010 to 87% in 2020. Between 2020 and 2019, VCRs declined in six countries (Bulgaria, Croatia, Estonia, Italy, Lithuania, and Poland). This trend continued in 2021 for the above-mentioned countries, with the exception of Italy and Poland, alongside the Czech Republic, Ireland, Latvia, Romania, and Spain.

### 3.4. Measles Vaccine

Coverage rates for measles 1st dose (Figure 5) were lower in 2019 than 2010 in thirteen countries (Bulgaria, Croatia, Cyprus, Czech Republic, Estonia, Finland, Greece, Lithuania, The Netherlands, Poland, Romania, Slovakia, and Slovenia). Coverage declined between 2019 and 2020 in eight countries (Bulgaria, Croatia, Denmark, Italy, Lithuania, Malta, Poland, and Romania). In 2021, coverage continued to decline compared to 2020 in the previous countries (except for Bulgaria, Denmark, and Italy) as well as in Estonia, Finland, Ireland, Latvia, The Netherlands, Slovakia, and Spain. Between 2020 and 2021, coverage fell in eight countries (Croatia. Estonia, Latvia, Lithuania, Malta, Slovakia, Spain, and Sweden), while it increased in four countries (Austria, Bulgaria, Denmark, and The Netherlands).

Similarly, coverage for measles 2nd dose was lower in 2019 than 2010 (Figure 6). The decline is particularly steep in Romania, where coverage dropped from 93% to 76% (in 2010 and 2019, respectively). Declining coverage continued in 2020, with ten countries experiencing drops compared to 2019 (Bulgaria, Croatia, Estonia, Ireland, Italy, Latvia, Lithuania, The Netherlands, Portugal, Romania, and Slovenia).

### 3.5. Polio Vaccine

Polio coverage rates follow a similar declining trend (Figure 7), with coverage lower in 2019 than 2010 in sixteen countries (Austria, Bulgaria, Croatia, Cyprus, Czech Republic, Estonia, Finland, France, Germany, Lithuania, The Netherlands, Poland, Romania, Slovakia, Slovenia, and Spain). Coverage rates were lower in six countries in 2020 compared to 2019 (Bulgaria, Finland, Lithuania, Romania, Spain, and Sweden). This trend continued, with coverage further declining in the same countries (except Sweden), alongside an additional five countries (Croatia, Czech Republic, Estonia, Germany, Latvia, and Slovenia), between 2020 and 2021. As of 2021, only twelve countries met the 95% target for the polio vaccination (Belgium, Cyprus, Denmark, France, Greece, Hungary, Luxembourg, Malta, The Netherlands, Portugal, Slovakia, and Sweden), five less than in 2010.

## 4. Discussion

### 4.1. Vulnerability of Vaccination Systems

Variations or fluctuations in VCRs are a concern. Overall, countries that have reported lower VCRs tend to have more and severer fluctuations in such coverage. Vaccine fluctuations highlight the fragility of vaccination coverages and of the immunisation ecosystem. Strong efforts are needed to ensure countries have robust and resilient immunisation systems to tackle this tendency.

#### 4.1.1. Vaccine Misinformation

Both insufficient vaccine coverages and fluctuations act as barriers to effective “herd immunity” and enhance outbreak risk. These barriers may be linked to vaccine hesitancy [16,17]. Misinformation, which often appears in tandem with vaccine hesitancy and the anti-vaccine movement, is highlighted as a reason for both Austria’s and Bulgaria’s suboptimal coverage rates [16]. According to a 2018 European Commission report, the Bulgarian population were found to be the least likely to agree that vaccines are safe [17]. Similarly, it is noted as a reason for Romania’s declining coverage rates over the past decade (e.g., HepB3 coverage fell from 98% in 2010 to 87% in 2020) [16,18]. This sentiment is echoed in the 2022 European Commission report; Austria, Belgium, Bulgaria, Croatia, Cyprus, Czech Republic, Denmark, Estonia, Finland, France, Germany, Greece, Hungary, Latvia, Lithuania, Malta, The Netherlands, Slovakia, Slovenia, Spain, and Sweden express a decrease in vaccine confidence regarding the State of Vaccine Confidence in the European Union [19]. To combat this, evidence suggests that implementing robust public awareness campaigns, alongside reliable medical advice, can be successful [17]. Involvement of health professionals has also been identified as crucial in driving vaccine confidence [20,21].

Awareness campaigns have proven to drive positive change on vaccine coverage. Bulgaria saw positive changes in VCRs after 2016, when an informational campaign about vaccines was launched [22]. The project provided information on the Bulgarian immunisation schedule, vaccine safety, and the benefits of immunisation. The development of a website that promoted community engagement and provided the possibility of communication between individuals and health professional was noted as one of the most effective tools [22]. However, whilst coverage increased following the 2016 campaign, it dropped in 2020 and 2021, which coincides with the COVID-19 pandemic. This corroborates the notion of increased immunisation programme vulnerability in times of crises.

However, vaccination awareness campaigns alone may be insufficient. In Austria, an awareness campaign was implemented in early 2014 to increase vaccination rates. It was coupled with a national action plan on measles, mumps, and rubella (MMR) elimination. This plan targeted refugees, given the high priority to immunise this population [16]. Whilst measles VCRs increased following the start of the campaign, there was a decline in 2018. However, coverage for vaccines that were not the focus of the campaign declined after 2014, signalling that the campaign failed to have a wider impact on attitudes to vaccination within Austria.

#### 4.1.2. Vaccine Policies

Vaccination policies vary among EU countries. Twelve EU countries (Belgium, Bulgaria, Croatia, Czech Republic, France, Hungary, Italy, Latvia, Malta, Poland, Slovakia, and Slovenia) have mandatory vaccination policies for at least one vaccine in the paediatric vaccination schedule [23]. However, vaccine mandates have various levels of success regarding coverage [24]. For example, in Italy, France, and Latvia it positively affected coverage. Yet, vaccination is mandatory in Bulgaria, and it experiences high degrees of fluctuation in VCRs, with coverage below the designated threshold for all vaccines assessed in 2021 [24]. Nonetheless, policies that have promoted regular health evaluations for infants, or the harmonisation of the immunisation schedule with routine health check-ups for children, have been a facilitator for increased coverage [25]. Some EU countries, such as Estonia and Germany, have successfully introduced mandatory check-ups for infants, where vaccines may be given albeit not mandatory themselves [25].

Policy changes pertaining to digital technologies has been shown to combat vaccine misinformation and increase VCRs: digital reminder systems increased awareness about correct registrations of vaccines in Denmark [26]. Data shows an increase in coverage for most vaccines following the introduction of a 2014 policy that centred on digital technologies [26]. It allowed the Danish Ministry of Health, through the Statens Serum Institut, to use the national electronic Immunisation Information System to send written reminders to parents with missing childhood vaccinations, as well as give access to an online overview of vaccination status’ or send reminders about missing vaccinations.

#### 4.1.3. Vaccine Accessibility

Increased vaccine accessibility means reducing structural barriers. Structural barriers hinder vaccination uptake, such as requiring taking time off work to take a child to the doctor [27]. In Romania, in 2015, the place of administration for school-age children was changed from school to the family doctor [28]. Subsequently, coverage fell in 2015 across many vaccines. Belgium has consistently high coverage rates for most vaccines, above 95% coverage, except for measles 2nd dose. Reasons for these rates are thought to include improved accessibility to vaccines; vaccination is provided through public health services and primary health care providers alongside paediatricians [16]. Services are geographically well-distributed across the country and are completely free of charge—requiring only an administration fee when carried out by the paediatrician [16]. An additional barrier is that of procurement delays [16,18]. In Romania, no DTP vaccine doses were purchased by the health ministry at the end of 2016 [28]. A corresponding decrease in coverage rates for DTP 3rd dose occurred in 2017.

An example of a reduction in these structural barriers is the hexavalent vaccine. The combination vaccine allows for reduced visits to a healthcare professional, as well as lower vaccination costs. They have also been associated with increases in coverage and more timely vaccination [29]. In Malta, the introduction of the hexavalent vaccine into the national childhood vaccination programme in 2010 was followed by an increase in vaccine coverage [23,30]. DTP 3rd dose coverage, which is a key indicator for routine immunisation programme performance [30], increased from 76% in 2010 to 96% in 2011, and remained over threshold levels for the subsequent years. In France, HepB3 coverage steadily increased. This progression began in 2008, which is when the hexavalent vaccine that includes HepB3 first became reimbursable [31].

### 4.2. Current Challenges: The Pandemic and the Ukraine Crisis

The COVID-19 pandemic has had an impact on vaccination services in many countries. This is seen in the trend of declining vaccination coverage across all vaccines in 2019 compared to 2010, with further reduction in 2020 and 2021 [15]. Data suggests pandemic disruption was higher in 2021; however, this could be a result of delayed data collection since coverage estimates do not occur in real time. Hib3 coverage dropped in 2021 compared to 2020 for 12 countries [15]. HepB3 coverage rates follow a similar trend; coverage declined in 2020, after the onset of the pandemic, with most of these countries experiencing further decline in 2021 [15]. Global 1st and 2nd dose measles coverage dropped, as well as polio vaccine coverage [10]. In France, 10.7% less MMR vaccines and 18.3% less tetanus vaccines were administered in March 2020 compared to the same period in previous years [32]. This decline continued into 2021. This trend was repeated in Greece, where a sharp decline in coverage across all vaccines was recorded between 2020 and 2021 [33]. VCRs fluctuated according to lockdown measures, with data suggesting almost zero vaccination of adolescents in February 2021, when the third lockdown unfolded. The impact of the pandemic on vaccination coverage varies across EU countries, likely due to the differences in containment measures implemented and how different countries were impacted by COVID-19. Vaccination schedule disruption during the pandemic was further exacerbated due to the resource diversion to COVID-19 relief efforts, including service and supply chain disruptions [10,11]. This drop in coverage rates increases the likelihood of VPD outbreaks; inadequate coverage levels have already resulted in avoidable measles and polio outbreaks [11,34].

VPD outbreaks are of particular importance given the ongoing Ukraine crisis. Ukraine and neighbouring EU countries, such as Poland and Romania, already struggle with immunity gaps in their population, particularly in the paediatric population [12]. These countries have below threshold coverage rates for the polio and both doses of measles vaccine. This places an additional burden on healthcare systems; existing immunity gaps in the refugee host countries will be especially vulnerable [35]. Ukraine has been experiencing an outbreak of polio since October 2021 and is endemic for measles [12]. Diphtheria is also a source of concern; cases may be exacerbated due to lack of access to water, sanitation, and hygiene, alongside suboptimal coverage for routine and childhood vaccinations [35]. Moreover, disruption of vaccination programmes without effective catch-up heighten the risk of VPDs. To counteract this, a series of actions could be implemented, such as enhanced outreach services, supplementary immunisation activity, and strengthening routine immunisation data systems [36]. Monitoring VPD outbreaks and ensuring systematic real-time data collection has been crucial to allow for early diagnosis and case management [37].

Table 1 summarises the enablers and barriers discussed here. Table 2 affords a brief overview of recommendations derived from the evidence reviewed. Recommendations focus on the areas of vaccine accessibility, vaccine information, and crisis plans. These recommendations are based on the World Federation of Public Health Associations (WFPHA) International Immunisation Policy Taskforce key recommendations to improve the resilience of paediatric vaccination programmes in the EU for policy makers to take forward at EU and national levels [38].

## 5. Conclusions

Over the past decade, VCRs have fluctuated greatly throughout the EU, with lower performing countries proving to be more susceptible to these fluctuations. There can be multiple factors behind insufficient VCRs in EU countries, with vaccine hesitancy considered among the key contributors. Countries implement a range of actions to try to improve coverage rates, such as public awareness campaigns or policy changes, that may correlate with increases in coverage. However, barriers related to vaccine accessibility (or lack thereof) and vaccine misinformation may be associated with a decrease in vaccine coverage rates. Evidence suggests facilitating access to vaccines has proven to increase coverage. This includes factors such as effective catch-up campaigns, access to immunisation services, public coverage of the costs of vaccinations, and information/education campaigns, which acted as levers to vaccination coverage rates. Moreover, with the current Covid-19 and Ukraine crises converging, the mass population movement poses a substantial risk of international spread due to gaps in vaccination coverage. It places an additional burden on healthcare systems. Countries impacted most by current challenges are more vulnerable to unexpected crises, which fuels the existing challenges related to reduced coverage rates. To ensure the best protection of all from preventable diseases, evidence recommends concrete crisis preparedness plans as well as constant action to achieve and maintain robust and resilient vaccination systems.

## Figures and Tables

**Figure 1 vaccines-11-00184-f001:**
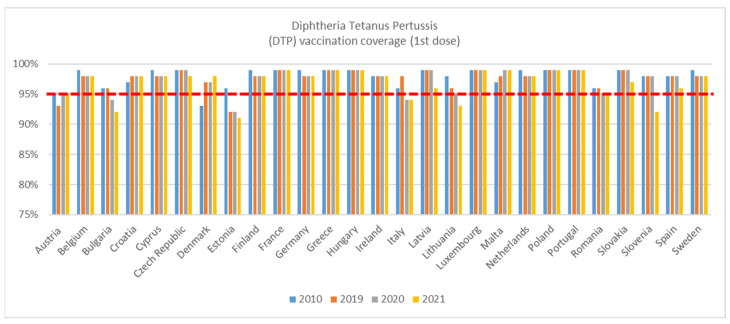
WHO, DTP 1st dose immunisation data—WUENIC, accessed 1 July 2022.

**Figure 2 vaccines-11-00184-f002:**
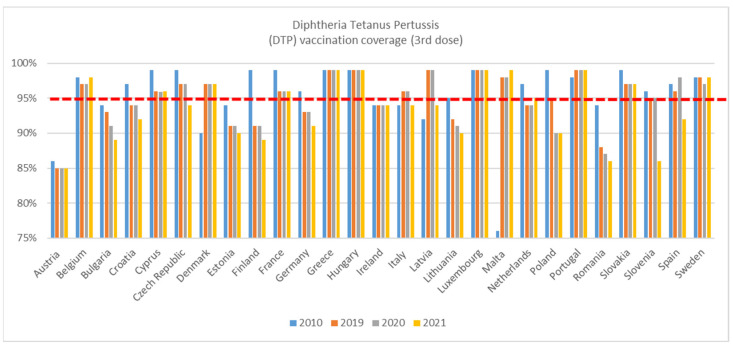
WHO, DTP 3rd dose immunisation data—WUENIC, accessed 1 July 2022.

**Figure 3 vaccines-11-00184-f003:**
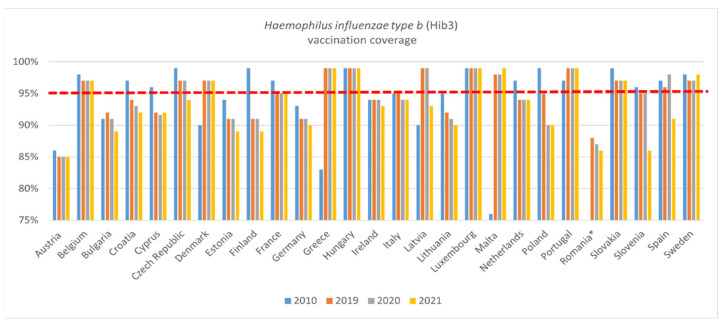
WHO, Hib3 immunisation data—WUENIC, accessed 1 July 2022. * 2010 data not available for Romania.

**Figure 4 vaccines-11-00184-f004:**
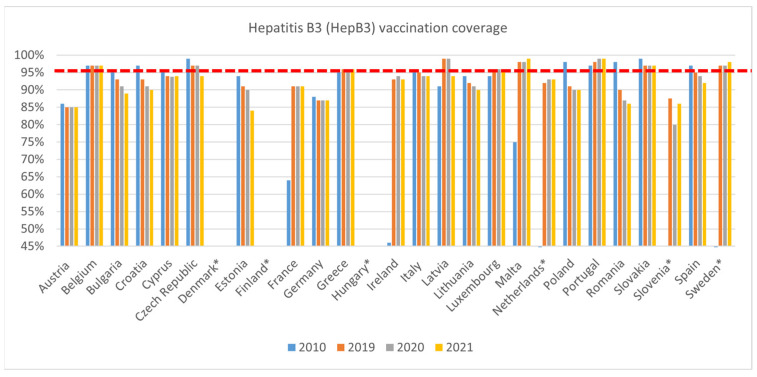
WHO, HepB3 immunisation data—WUENIC, accessed 1 July 2022. * The Netherlands: a different source was used for 2010 data: WHO, immunisation data—OFFICIAL, accessed 1 June 2022; Slovenia: a different source was used for 2019 and 2020 data: WHO, immunisation data—OFFICIAL, accessed June 2022; Sweden: a different source was used for 2010 data: Swedish Institute for Infectious Disease Control, Vaccinationsstatistik från barnavårdscentralerna, insamlad januari 2010, gällande barn födda 2007, 2010; 2010 data not available for Slovenia; No data available for Denmark, Finland, and Hungary—these countries do not include this vaccine in the national immunisation programme.

**Figure 5 vaccines-11-00184-f005:**
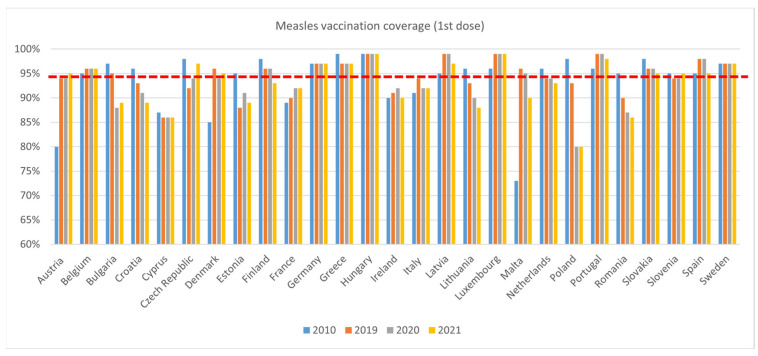
WHO, Measles 1st dose immunisation data—WUENIC, accessed 1 July 2022.

**Figure 6 vaccines-11-00184-f006:**
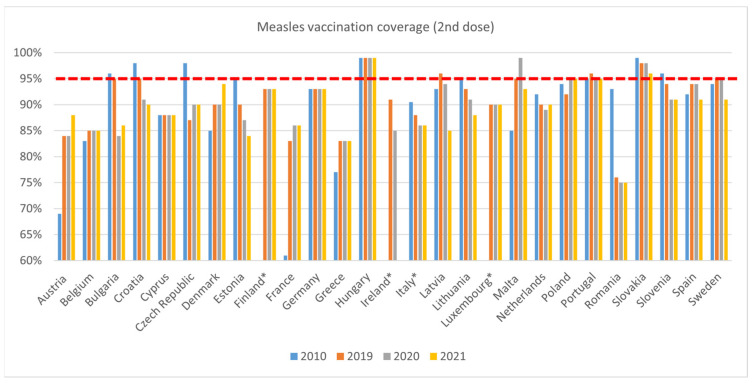
WHO, Measles 2nd dose immunisation data—WUENIC, accessed 1 July 2022. * Italy: a different source was used for 2010–2012 data: Ministero della salute, Vaccinazioni dell’età pediatrica—Anno 2010, accessed 1 June 2022; Finland: a different source was used for 2011–2013 data: Finnish Institute for Health and Welfare, THL, Vaccination coverage in children, accessed 1 June, 2022; 2010, 2014, and 2015 data not available for Finland; 2010 and 2011 data not available for Luxembourg; Ireland: No WHO data available for 2010 and 2021. A different source was used for 2011–2020 based on the school year, usually running from August/September until June, Health Protection Surveillance Centre, accessed 1 July 2022.

**Figure 7 vaccines-11-00184-f007:**
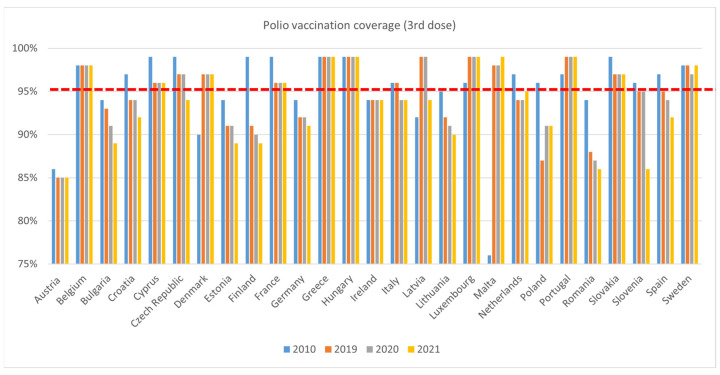
WHO, Polio 3rd dose immunisation data—WUENIC, accessed 1 July 2022.

**Table 1 vaccines-11-00184-t001:** Summary of enablers and barriers to VCRs.

Barriers	Enablers
Structural barriers (e.g., having to take time off work to get a child vaccinated) [27]	Easy access to vaccine administration [16,39]
Delays in vaccine procurement [16,40,41]	Outreach services and catch-up campaigns [16,42]
Vaccine hesitancy (among public and/or health care professional) [16,40]	Vaccination policies (depending on local context) [24,40,43,44]
Spread of misinformation about vaccination [16]	Real-time systematic data collection [37]
	Communication between individuals and healthcare workers regarding vaccines [16,22]
	Sending vaccine reminders to parents [26]
	Awareness raising and education campaigns [16]
	Easily accessible and reliable information materials about vaccination [22,45]

**Table 2 vaccines-11-00184-t002:** Brief overview of recommendations.

Recommendations	
General	Specific
Facilitate access to vaccination	Increase range of providers to administer vaccination
	Enable providers to administer vaccinations outside of the site where they are employed (e.g., administer vaccines at childcare facilities)
	Additional hours to access vaccinations (e.g., outside of regular work hours)
Develop mitigation strategies and plans to respond to VPD outbreak	Ensure sufficient infrastructure and health worker capacity
Increase use of digital technologies	Improve real-time data collection and disease surveillance systems
	Electronic immunisation records
	Automated vaccination reminders
Improve education and awareness regarding the value of vaccination	Public awareness and communication campaigns
	Public health education programmes

## Data Availability

Analysis of primary data extracted from externally available datasets was conducted in July 2022. The analysis included vaccination coverage data in all 27 EU Member States between 2010 and 2021. All vaccination coverage data in this brief is based on the WHO/UNICEF Joint Estimates of National Immunisation Coverage (WUENIC) data, unless otherwise indicated. Where WUENIC data was unavailable, other comparable data was used, based on other available WHO data or official national datasets, such as from national Ministries of Health. WUENIC (WHO/UNICEF), Immunisation data; The Netherlands HepB vaccination coverage, 2010 source: WHO, Immunisation data—OFFICIAL; Slovenia HepB vaccination coverage: a different source was used for 2019 and 2020 data: WHO, Immunisation data—OFFICIAL; Sweden HepB vaccination coverage: a different source was used for 2010 data: Swedish Institute for Infectious Disease Control, Vaccinationsstatistik från barnavårdscentralerna, insamlad januari 2010, gällande barn födda 2007, 2010; Italy measles 2nd dose vaccination coverage data: a different source was used for 2010–2012 data: Ministero della salute, Vaccinazioni dell’età pediatrica—Anno 2010; Finland: a different source was used for 2011–2013 data: Finnish Institute for Health and Welfare, THL, Vaccination coverage in children; Ireland measles 2nd dose vaccination coverage data, 2011–2020 based on school year, Health Protection Surveillance Centre.

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
