# Peer review of "Driving Paediatric Vaccine Recovery in Europe"

_vaccines, 2023, doi:10.3390/vaccines11010184_

Round 1

Reviewer 1 Report

This manuscript is a review about paediatric vaccination in Europe. In general, the study is well described. The authors described, that paediatric vaccination declined in the period between 2010 to 2021. Thus, this paper is very informative and focus on a trend (declining vaccination coverage rates), which could become a challenge in the next years, causing economic and disease burden. However, this paper needs to be improved before it can be published: 

-        As a review, this manuscript should also discuss deeper the influence of the COVID pandemics in the vaccine fluctuation between the years 2010-2021. It is expected a difference in the numbers between pandemic years and non-pandemic years (lines 44-46: “the disruption of routine immunization”). The authors should discuss in more detail not only the point 4.2. (line 286), but also the declining VCR before the pandemics and conflict in Ukraine (e.g. measles between 2017-2018). Why the campaign in Austria failed to have a wider impact on vaccination (line 242)?

To drive positive change on vaccine coverage, is it better to explain and convince the population about the benefits of vaccination then to obligate the vaccination? Please, discuss in more detail the numbers between countries with and without mandatory vaccination policies.

-        Line 303: “… in their population., particularly in…”

Please, remove the point between “population” and “, particularly”.

-        In the conclusion the authors explain how important are the accessibility and information regarding vaccines. Yes, that is a key point. Therefore, to encourage the population to protect their children by vaccination, the authors should better highlight the benefits of vaccines. The authors should explain, that the goal of vaccines is not necessarily always to prevent disease, but to save lives. Successful vaccination protects against death. Even if vaccinated people get sick, this not means that the vaccine did not work. The main goals are to avoid hospitalization and death. The COVID-pandemic was an example, how crucial is the communication between experts, politicians and the population, to reach and convince as many people as possible, how important are vaccines. This deeper discussion would impact in the significance of this review.

Author Response

Many thanks for your review. Answers to your revisions are provided below:

As a review, this manuscript should also discuss deeper the influence of the COVID pandemics in the vaccine fluctuation between the years 2010-2021. It is expected a difference in the numbers between pandemic years and non-pandemic years (lines 44-46: “the disruption of routine immunization”). The authors should discuss in more detail not only the point 4.2. (line 286), but also the declining VCR before the pandemics and conflict in Ukraine (e.g. measles between 2017-2018). Why the campaign in Austria failed to have a wider impact on vaccination (line 242)? 

  • We agree more detail is required when discussing the impact of COVID-19 pandemic. In the introduction, a statement highlighting the impact of the pandemic on VCRs has been included. 
  •  Point 4.2. Current Challenges: We agree; an expansion of the discussion regarding the impact of COVID-19 on schedule disruption has been included. We opted to not expand the topic of measles 2017-2018 given that this would require its own section given the importance (i.e., a case study of sorts), and believe that we go sufficiently in depth into the declining VCR prior to the current challenges.
  • Did not expand on Austria (line 242), as the declining VCR serve to signal failure of campaign to have wider impact.

To drive positive change on vaccine coverage, is it better to explain and convince the population about the benefits of vaccination then to obligate the vaccination? Please, discuss in more detail the numbers between countries with and without mandatory vaccination policies.

  • Point 4.1.2. Vaccine policies: We expanded the discussion of mandatory vaccine policies to include the twelve EU countries, as well as making note of countries where other vaccine policies have fomented vaccine uptake

-        Line 303: “… in their population., particularly in…”; Please, remove the point between “population” and “, particularly”.

  • Removed 

-        In the conclusion the authors explain how important the accessibility and information regarding vaccines are. Yes, that is a key point. Therefore, to encourage the population to protect their children by vaccination, the authors should better highlight the benefits of vaccines. The authors should explain, that the goal of vaccines is not necessarily always to prevent disease, but to save lives. Successful vaccination protects against death. Even if vaccinated people get sick, this not means that the vaccine did not work. The main goals are to avoid hospitalization and death. The COVID-pandemic was an example, how crucial is the communication between experts, politicians and the population, to reach and convince as many people as possible, how important are vaccines. This deeper discussion would impact in the significance of this review.

  • Point 1. Introduction: We agree with reviewer. Albeit having mentioned this in the introduction, it has been expanded upon to clearly highlight the importance of paediatric vaccination. 

Reviewer 2 Report

Abstract. Line 8-9. Please rephrase it as it does not make any sense.

Methods. I'd list, for each country, all websites/data source investigated (as supplementary material).

References. Ref number 25 and 44 are the same. A good replace for reference number 25 describing measles elimination in Italy is: Adamo G, et al. Towards elimination of measles and rubella in Italy: Progress and challenges. PLoS One. 2019 Dec 16;14(12):e0226513.

Author Response

Many thanks for your review. Answers to your revisions are provided below:

Abstract. Line 8-9. Please rephrase it as it does not make any sense.

  • Abstract: rephrased it such that it now reads: Childhood vaccination coverage has increased throughout Europe in the past decades. However, challenges persist in many areas within the EU resulting in declining coverage rates in many countries in the period between 2010 and 2021.

Methods. I'd list, for each country, all websites/data source investigated (as supplementary material).

  • Although this would be ideal, it is quite an arduous task given that many sources often cover various countries. In the data availability statement we highlight how majority of data (unless otherwise stated) was obtained from externally available datasets (WUENIC data). Other data sources are clearly mentioned there, per country per year (e.g., Slovenia HepB vaccination coverage source for 2019 and 2020 was different) – and have afforded links to this data.

References. Ref number 25 and 44 are the same. A good replace for reference number 25 describing measles elimination in Italy is: Adamo G, et al. Towards elimination of measles and rubella in Italy: Progress and challenges. PLoS One. 2019 Dec 16;14(12):e0226513.

  • While the article mentioned is a good replacement reference, to ensure continuity of the article we have opted to simply remove the duplicate.